# A Multilevel Guidance-Exploration Network and Behavior-Scene Matching Method for Human Behavior Anomaly Detection

## ABSTRACT

Human behavior anomaly detection aims to identify unusual human actions, playing a crucial role in intelligent surveillance and other areas. The current mainstream methods still adopt reconstruction or future frame prediction techniques. However, reconstructing or predicting low-level pixel features easily enables the network to achieve overly strong generalization ability, allowing anomalies to be reconstructed or predicted as effectively as normal data. Different from their methods, inspired by the Student-Teacher Network, we propose a novel framework called the Multilevel Guidance-Exploration Network (MGENet), which detects anomalies through the difference in high-level representation between the Guidance and Exploration network. Specifically, we first utilize the Normalizing Flow that takes skeletal keypoints as input to guide an RGB encoder, which takes unmasked RGB frames as input, to explore latent motion features. Then, the RGB encoder guides the mask encoder, which takes masked RGB frames as input, to explore the latent appearance feature. Additionally, we design a Behavior-Scene Matching Module (BSMM) to detect scene-related behavioral anomalies. Extensive experiments demonstrate that our proposed method achieves state-of-the-art performance on ShanghaiTech and UB-normal datasets, with AUC of 86.9 % and 74.3 %, respectively. The code is available on GitHub.

## CCS CONCEPTS

• **Computing methodologies** → **Scene anomaly detection**.

## KEYWORDS

human anomaly detection,one-class, multimodal features.

## 1 INTRODUCTION

Human behavior anomaly detection aims to temporally or spatially localize the abnormal actions of the person within a video. It plays a significant role in enhancing public security [32, 41]. Detecting such anomalies presents a challenge due to the infrequent occurrence and the various types of abnormal events [47]. As a result, most typical methods [7, 17, 22, 27, 36, 46], employ **one-class** methods using only normal data for training (also referred as **unsupervised** learning methods in this area). In these approaches, including our method, behaviors that the model identifies as outliers are considered anomalies.

**Unpublished working draft. Not for distribution.**

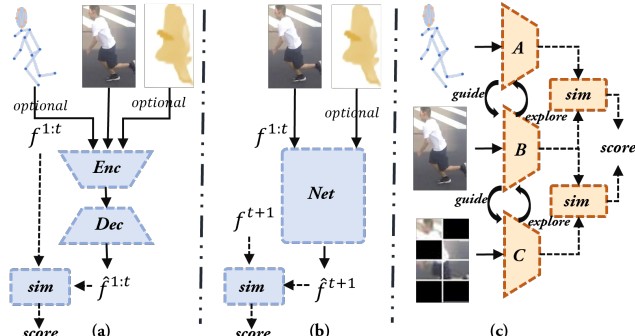

**Figure 1: Comparison of different methods using various features. (a) Reconstruction-based method, using the autoencoder to reconstruct the previous $T$ frames $f^{1:t}$. (b) Prediction-based method, predicting the $t + 1$ frame $f^{t+1}$ from the prior $T$ frames. Both of them detect anomalies based on reconstruction or prediction errors. (c) Our Multilevel Guidance-Exploration framework, includes two similar levels. For instance, in the 1-st level, Encoder-B learns another type of feature under the guidance of a pre-trained network (Encoder-A), detecting anomalies based on the similarity of latent output features.**

Among these unsupervised learning approaches, many methods use reconstruction or frame prediction methods combined with various features to detect human behavior anomalies. The reconstruction-based framework[15, 23, 27, 28, 39], illustrated in Figure 1(a), utilizes autoencoders trained on normal data, detecting anomalies based on elevated reconstruction errors. For example, Wang et al. [39] propose a new autoencoder model, named Spatio-Temporal Auto-Trans-Encoder, to enhance consecutive frame reconstruction. The prediction-based methods[4, 7, 18, 22, 23, 46], as depicted in Figure 1(b), typically predict pixel-level features for the next frame using previous frames. Foe example, Cai et al. [7] propose an Appearance-Motion Memory Consistency Network based on autoencoders, explicitly considering the endogenous consistency semantics between optical flow features and RGB appearance features during the prediction process. Additionally, Liu et al. [23] propose a hybrid strategy by initially reconstructing optical flow features with a reconstruction autoencoder and then jointly predicting the next frame with previous frames,

However, reconstructing or predicting pixel-level features at a low level can result in the network having overly strong generalization [7, 27, 36, 46], where some anomalous samples can be reconstructed or predicted as effectively as normal samples. This phenomenon poses a challenge in distinguishing between normal and anomalous instances. Additionally, these approaches [4, 15–18, 23, 36] ignore scene context. They focus solely on the behavior

of individuals without considering their interaction with the surrounding scene. For example, lying on a zebra-crossing road should be considered anomalous compared to the same posture on a beach.

Different from the aforementioned method and inspired by teacher-student framework, we design a novel framework named Multilevel Guidance Exploration Network (MGENet), which focuses on exploring high-level feature difference rather than recovering or predicting pixel-level information. As shown in Figure. 1(c), MGENet detects motion and appearance anomalies using a Two-level guidance-exploration pattern. Each level is trained with different types of input features and leverages the difference in output features between the Guidance and Exploration Networks to detect anomalies. The principle behind this is that the model structures and input data formats differ between guidance and exploration networks. As a result, they might show distribution shifts towards unfamiliar anomaly types compared to normal patterns. The direction of these shifts is hard to consistently pinpoint, leading to larger difference compared to normal patterns.

Specifically, we employ Spatio-temporal Normalizing Flow [17] to map normal human-pose data into a latent representation characterized by a Gaussian distribution. This process strategically situates anomalous pose data at the distribution's periphery. Then, guided by Normalizing Flow, the RGB Encoder captures spatio-temporal features, detecting motion anomalies by analyzing the difference in the output features between these two networks. Furthermore, the RGB Encoder also guides the unmask Encoder to distill high-level features from specific patches of masked RGB frames, detecting appearance anomalies based on the similarity between the high-level features output by both networks. Additionally, we incorporate a Behavior-Scene matching module, which establishes and stores the relationship between normal behavior and scenes, enabling the detection of scene anomalies. Finally, we demonstrate the effectiveness of the method on two publicly available datasets.

In summary, the contributions of our work are as follows:

- We design a multilevel guidance-exploration pattern, similar to a teacher-student network, but both of two networks are not merely knowledge distillation, but rather collaboratively detect anomalies.
- We propose a behavior-scene matching method to store the relationship between scenes and behaviors and can detect scene-related anomalies.
- Different from reconstruction-based or prediction-based methods, we detect motion and appearance anomalies based on the high-level feature difference between two levels of guidance and exploration networks, and detect scene-related anomalies based on the disparity in the matching degree between behaviors and scenes.
- Extensive experiments demonstrate that our proposed method outperforms existing unsupervised methods, achieving a state-of-the-art AUC of 86.9% on the ShanghaiTech dataset and 74.3% on the UBnormal dataset.

## 2 RELATED WORK

### 2.1 Video Anomaly Detection

In recent years, numerous studies have achieved remarkable results based on RGB frames, optical flow, or pose features.

Some researchers employ reconstruction methods[15, 23, 27, 28, 32], for anomaly detection, assessing anomalies based on higher reconstruction errors compared to normal samples. Park et al. [27] propose augmenting the autoencoder with a memory module, favoring proximity to normal samples during reconstruction and amplifying errors for anomalies. Sun and Gong [32] utilize two autoencoders to reconstruct motion and appearance features. Furthermore, they also design a contrastive learning method to identify scene-related behavioral anomalies, but they only detect within limited scenarios, lacking diversity. Ristea et al. [28] integrates reconstruction functionality into a novel self-supervised predictive building block, trained to predict masked information in a self-supervised manner. Some researchers use prediction methods[4, 7, 18, 22, 23, 46], to detect anomalies. Liu et al. [22] propose a future frame prediction approach, which detects anomalies by assessing the discrepancies between predicted images and actual images. Chen et al. [9] find limitations in simple prediction constraints for representing appearance and flow features. They introduce a novel bidirectional architecture with three consistency constraints to regulate the prediction task. Yang et al. [46] introduced the task of key frame restoration, encouraging Deep Neural Networks to infer missing frames based on video key frames, thereby restoring the video.

Furthermore, in recent years, there has been an emergence of utilizing alternative methods for anomaly detection. Hirschorn et al. [17] employ Normalizing Flow to map normal data into the latent representation, locating anomalous data at the distribution periphery. Wang et al. [36] propose a new pretext task, disrupting both temporal and spatial order and training the model to restore RGB frames.

### 2.2 Student Teacher Network

Initially, the student-teacher network was applied to knowledge distillation[41]. The student model, which has fewer parameters, learns from the output of the teacher model with a larger parameter size, enabling the student model to achieve performance that closely approximates that of the teacher model. Recently, in the field of industrial anomaly detection[5, 21, 29, 44], Bergmann et al. [5] are the first to propose an unsupervised anomaly detection framework based on teacher-student learning. And some researchers [29] employ knowledge distillation to detect anomalies by utilizing the regression error of student networks on the feature outputs of a high-parameter teacher network. The STPM method [44] is based on student-teacher feature pyramid matching, with the student and teacher networks being pre-trained as ResNet50 and ResNet18, respectively.

The methods mentioned earlier and our approach's appearance anomaly detection phase share similarities, employing knowledge distillation. However, there's a difference in the motion anomaly detection process, where knowledge distillation is not the primary emphasis. The Normalizing Flow has a lightweight architecture, whereas the RGB Encoder has a more complex structure. Additionally, there are substantial differences in the frameworks and input data types between these two networks.

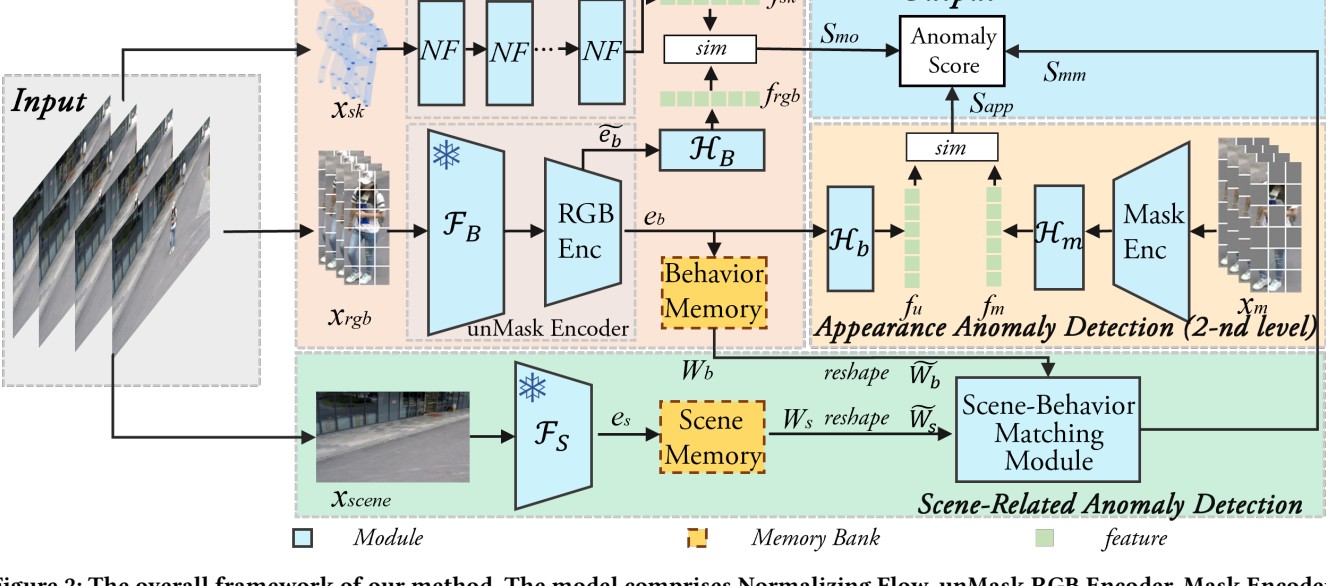

**Figure 2: The overall framework of our method. The model comprises Normalizing Flow, unMask RGB Encoder, Mask Encoder, and Scene-Behavior Matching Module. Normalizing Flow serves as guidance network at the 1-st level, while unMask RGB Encoder serves as both the 1-st level's exploration network and the 2-nd level's guidance network, with Mask Encoder acting as the 2-nd level's exploration network. $\mathcal{F}_B$ and $\mathcal{F}_S$ are frozen feature extractors.**

## 2.3 Masked Visual Model

Masked Visual Modeling [31, 40, 48] improves visual representation learning by masking image portions. Chen et al. [10] propose a pretraining method with two tasks: predicting representations for masked patches and reconstructing masked patches. Zhang et al. [49] demonstrate improved performance with only supervised visible patches, omitting the need for masked patches. In the realm of video representation learning, Tong et al. [35] show that video-masked autoencoders are also data-efficient learners for self-supervised video pre-training. Wang et al. [37] present Masked Video Distillation, a succinct two-stage framework, for video representation learning. Inspired by masking tasks, we mask partially video frames and exclusively use visible frames to learn the latent high-level features of the uncovered frames. In this way, since the model has not encountered the appearance of anomalies before, there will be differences in the latent representation of unmasked frames containing anomalies.

## 3 METHOD

### 3.1 Overview

Figure 2 illustrates the overall framework. Given a video clip with $T$ consecutive frames, we extract human-centric RGB frames $x_{rgb} \in \mathbb{R}^{T \times H \times W \times C}$ and $V$-joints skeletal pose data $x_{sk} \in \mathbb{R}^{T \times V \times C}$ [12, 43]. Meanwhile, according to BEIT [3], we segment the RGB frames into $P$ patches and mask the rate of $\gamma$ of the patches to obtain $x_m$. Then, anomalies are detected through the following four processes.

**Motion Anomaly Detection:** First, we pre-train Normalizing Flow to project $x_{sk}$ into a latent representation $f_{sk} \in \mathbb{R}^{T \times C_{mo}}$ following the Gaussian distribution. Then, it guides the RGB Encoder

and head $\mathcal{H}_B$ using RGB frames to learn spatio-temporal features $f_{rgb} \in \mathbb{R}^{T \times C_{mo}}$.

**Appearance Anomaly Detection:** We use head $\mathcal{H}_b$ to map behavior Features $e_b$ into $f_u \in \mathbb{R}^{T \times C_{app}}$. Following this, it guides the Mask Encoder using masked RGB frames $x_m$ to learn appearance features $f_m \in \mathbb{R}^{T \times C_{app}}$.

**Scene-related Anomaly Detection:** The scene undergoes feature extraction $\mathcal{F}_S$ to generate scene features $e_s$, alongside corresponding behavior features $e_b$, pass through the memory banks to get soft addressing weights $W_b \in \mathbb{R}^{P \times N_b}$ and $W_s \in \mathbb{R}^{1 \times N_s}$, respectively. Here, $N_b$ and $N_s$ represent the number of slots in the Behavior and Scene Memory, respectively. Then, they pass into the Behavior-Scene Matching Module together to compute the scene-related anomaly score $S_{mm}$.

**Anomaly Score:** The score is computed by considering the difference between the pose feature $f_{sk}$ and the behavior feature $f_{rgb}$, the similarity of appearance features $f_u$ and $f_m$, and the matching score $S_{mm}$.

### 3.2 Motion Anomaly Detection

Skeletal data helps the model capture the essential characteristics of movements or postures[19, 45]. Recently, Hirschorn et al.[17] designed the Spatio-temporal Normalizing Flow($NF_s$), including $L$ flow modules, which can map the skeletal distribution of normal skeletal data to a standard distribution through a series of invertible transformations, with anomalies typically found at the distribution's periphery.

To further distinguish between normal and abnormal samples, we use a Normalizing Flow with skeletal pose information as a

guidance network to assist the exploration network in learning features from RGB video frames. Firstly, we train Spatio-temporal Normalizing Flow according to [17], mapping pose data $x_{sk}$ to latent behavior features $f_{sk}$ :

$$f_{sk} = NF_s(x_{sk}). \tag{1}$$

Secondly, we use it as a pre-trained model to guide the exploration network (RGB Encoder) to generate the latent motion features $f_{rgb}$. The following are the detailed steps:

First, given RGB frames $x_{rgb} \in \mathbb{R}^{T \times H \times W \times C}$, similar to cube embedding [2, 11, 35], we treat each cube of $2 \times 8 \times 8$ as one token embedding, and obtain $t \times h \times w$ 3D tokens,where $t = \frac{T}{2}$, $h = \frac{H}{8}$, $w = \frac{W}{8}$. Then, map each token to the channel dimension. Next, we pre-extract RGB features of these tokens and employ the RGB Encoder, a small ViT backbone with joint space-time attention [11, 35], to obtain spatio-temporal features $e_b \in \mathbb{R}^{P \times C_b}$, where $P = t \cdot h \cdot w$.

Then, we reshape $e_b$ into $\tilde{e}_b \in \mathbb{R}^{t \times C_b \times h \times w}$ and design the spatial-temporal head $\mathcal{H}_B$, which replaces the $3 \times 3 \times 3$ convolution in Spatial-Temporal Excitation [8] with the decomposed Large kernel Attention, named large Spatial-Temporal Attention(LSTA), to further capture the spatio-temporal relationships of patches with long-distance temporal dependencies and spatial variations in different frames of human actions.

In detail,spatio-temporal head $\mathcal{H}_B$ consists of $L$ LSTA modules and the MLP layers. As shown in Figure 3, given an input tensor $e_b^{in} \in \mathbb{R}^{t \times c \times h \times w}$, we begin by performing channel-wise averaging, yielding a global spatio-temporal tensor $f \in \mathbb{R}^{t \times 1 \times h \times w}$. Then, we reshape $f$ into $f^* \in \mathbb{R}^{1 \times t \times h \times w}$ and pass it through the 3DLKA module to get transformed tensor $f_o^* \in \mathbb{R}^{1 \times t \times h \times w}$,which is represented as follows :

$$f_o^* = 3DLKA(f^*) = CONV(DWDC(DWC(f^*))), \tag{2}$$

where $DWC$ denotes a $\frac{k}{d} \times \frac{k}{d} \times \frac{k}{d}$ deep dilated convolution with dilated $d$ ,$DWDC$ denotes a $(2d-1) \times (2d-1) \times (2d-1)$ deep convolution, and $CONV$ denotes a $1 \times 1 \times 1$ convolution. Subsequently, $f_o^*$ is reshaped to $f_o \in \mathbb{R}^{t \times 1 \times h \times w}$ and passed through a sigmoid activation function to obtain the attention map. Finally, we use this map to guide $e_b^o$ for obtaining behavior feature $e_b^o$:

$$e_b^o = e_b^{in} + e_b^{in} \odot sigmoid(f_o), \tag{3}$$

where $\odot$ denotes the element-wise product. After passing through the MLP layers, we obtain the motion feature $f_{rgb}$. Finally, we minimize the difference between $f_{sk}$ and $f_{rgb}$ feature to facilitate the model in learning spatio-temporal pose features of normal patterns.

$$\mathcal{L}_{mo} = ||f_{sk} - f_{rgb}||_2^2. \tag{4}$$

In this way, for anomalous samples, achieving similar high-level semantic representation is more challenging due to differences in feature modalities and network architectures. Therefore, we can detect action anomalies based on the difference between the two types of features.

## 3.3 Appearance Anomaly Detection

Beyond motion anomalies, our method considers appearance anomalies, including carrying unidentified objects or using inappropriate

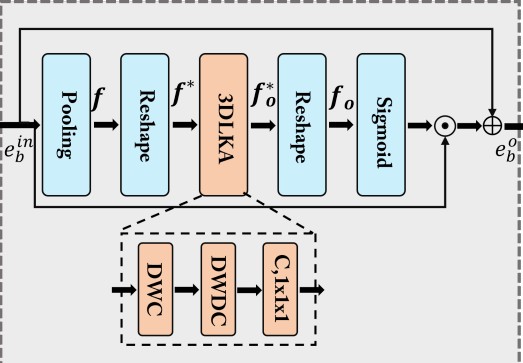

**Figure 3: The framework of LSTA.**

vehicles. Given this context, we extend Masked Image Modeling (MIM) [42] to video anomaly detection, enabling the Mask Encoder to learn normal appearance features guided by the unmasked RGB Encoder. This adaptation tackles challenges faced by the Mask Encoder in capturing high-level features of patches that were not encountered before but now are masked, resulting in noticeable differences from unmasked RGB features.

Specifically, following the approach of BEIT *MASK* [3], we mask patches with a ratio $\gamma$, which is set to 50% and obtain masked RGB frames $x_m$. Noted that we used the same mask for the frames within the same video frame, preventing the model from extracting patch features from adjacent frames.

$$\{x_m^1, x_m^2, ..., x_m^{N/2}\} = MASK\{x_{rgb}^1, x_{rgb}^2, ..., x_{rgb}^N\}, \tag{5}$$

where $N$ denotes the number of patches. Next, we also use the cube embedding method described in Section 3.2 to obtain tokens. These tokens pass through the Mask Encoder, which has a structure similar to the RGB Encoder, to learn appearance features. Then, the projection head is employed to obtain latent appearance features, denoted as $f_m$. Simultaneously, the appearance features $f_b$ also undergo the projection head to obtain latent appearance features $f_u$ with the same size as $f_b$.

During the training process, the Mask Encoder distills the high-level feature representation of the RGB Encoder under the condition of having only partially visible patches. The final loss function can be expressed as:

$$\mathcal{L}_{app} = 1 - \frac{f_m \cdot f_u}{||f_m|| \cdot ||f_u||}. \tag{6}$$

## 3.4 Scene-Related Anomaly Detection

**Formally, we postulate that unobserved behaviors within a scene should be categorized as anomalies.** To detect these anomalies, we design the Scene-Behavior Matching Module to capture the relationships between normal patterns of scenes and behaviors. As a result, scene-related behavior anomalies exhibit weaker matching with the learned features, leading to higher anomaly scores.

As illustrated in Figure. 4(a), the Behavior-Scene Matching Memory(BSMM), similar to the Behavior Memory and Scene Memory, is a read-write memory with a similar structure [27]. However, the key difference is that the Behavior-Scene Matching Memory

**Figure 4: Calculation process of (a) scene-related anomaly score and (b) appearance anomaly score. Here, S represents similarity calculation,*MASK* and $\overline{MASK}$ represent mutually opposite masks. Note that in figure(b), the two sets of masked images are sequentially processed through the Mask Encoder**

stores the representation of the addressing weights in the Behavior Memory and Scene Memory for the behavioral features and their corresponding scene features of all normal data. Below is an introduction to its update and read processes.

First, we choose a frame with the fewest individuals in a video as the scene image. Then, the scene image passes through the feature extractor, generating scene features $\mathbf{e}^s \in \mathbb{R}^{C_s}$. Then, behavior features $\mathbf{e}^b \in \mathbb{R}^{P \times C_b}$ and the corresponding $\mathbf{e}^s$ query the Behavior Memory and Scene Memory, respectively, and then similar to Eq.7, we contribute to calculating the similarity weights $W_b \in \mathbb{R}^{P \times N_b}$ and $W_s \in \mathbb{R}^{1 \times N_s}$, where $N_b$ and $N_s$ denote as the number of the slots in behavior memory and scene memory, respectively. In detail,For the $i$-th slot in behavior memory, denoted as $\mathbf{m}_i^b \in \mathbf{M}^b$, we can calculate the addressing weights between it and the $p$-th query $\mathbf{e}_p^b$ item as follows:

$$w_{i,p}^b = \frac{\exp\left(d\left(\mathbf{e}_p^b, \mathbf{m}_i^b\right)\right)}{\sum_{j=1}^{N_b} \exp\left(d\left(\mathbf{e}_p^b, \mathbf{m}_j^b\right)\right)}, \qquad (7)$$

where $d(\mathbf{e}^*, \mathbf{m}^*)$ denotes cosine similarity. The computation method for $w_i^s \in W^s$ is the same as that for $w_{i,p}^b$.

**Update:** We reshape $W^b$ and $W^s$ into one-dimensional vectors $\tilde{W}^b \in \mathbb{R}^{L_b}$ and $\tilde{W}^s \in \mathbb{R}^{L_s}$ ,where $L_b = P \times C_b$ and $L_s = 1 \times C_s$. Then, concatenate $\tilde{W}^b$ and $\tilde{W}^s$ along the channel:

$$\mathbf{W}^r = [\tilde{W}^b, \tilde{W}^s]. \qquad (8)$$

Next, similar to [27, 32], for each items $\mathbf{m}_i^r \in \mathbf{M}^r$ in the Behavior-Scene Memory, we update as following:

$$\mathbf{m}_i^r \leftarrow f\left(\mathbf{m}_i^r + \sum_{v \in U^i} v^{p,i} \mathbf{W}_v^r\right), \qquad (9)$$

where $f(\cdot)$ is the $L_2$ norm. $U^i$ represents the set of indices for the corresponding queries for the $i$-th item in the memory. $v^{p,i}$ represents matching probability between memory items and queries, similar to equation (7). It is worth noting that the aforementioned update operation occurs only in the final round and keeps the parameters of the Behavior Memory and Scene Memory unchanged.

**Read:** We calculate matching weights $C^*$ between similarity weights of behavior feature $W^b$ or scene feature $W^s$ and the Behavior-Scene Memory $\mathbf{M}^r$. Specifically, for the $i$-th slot in the Behavior-Scene Memory, the behavior matching weights $c_i^b \in C^b$ are calculated between $W_b$ and the first $L_b$ channels of the Matching Memory $\mathbf{M}^r$, as follows:

$$c_i^b = \frac{\exp\left(d\left(W^b, \mathbf{m}_{i,:L_b}^r\right)\right)}{\sum_{j=1}^N \exp\left(d\left(W^b, \mathbf{m}_{j,:L_b}^r\right)\right)}. \qquad (10)$$

Next, due to the sparsity of Memory, to avoid interference from irrelevant information dissimilar to behavior, we only select the Top-K items that most similar to behavioral features to calculate matching weights $C^b \in \mathbb{R}^K$. Similarly, we can use the above method with the last $L_s$ channels of $\mathbf{M}^r$ and the selected Top-K items to calculate $C^s$(Attach document have a more detailed procedure).

In this way, $W^b$ and $W^s$ serve as vector representations of historical behaviors and scenes. Their combination stored in $\mathbf{M}^r$ forms a pattern of the behavior-scene pattern. During the test phase, $W^b$ and $W^s$ act as query terms, individually computed with the behavioral and scene representations of each item $\mathbf{m}_i^r$ in $\mathbf{M}^r$ to derive $c_i^b$ and $c_i^s$. If the difference between them is significant, it indicates a mismatch between the behavior and the current scene. Finally, anomalies are measured by considering all patterns stored in the Matching Memory.

### 3.5 Loss Function and Anomaly Score

**Loss Function:** The training loss includes the regression loss $\mathcal{L}_{mo}$, and the distillation loss $\mathcal{L}_{app}$. Additionally, to allocate similar queries to the same item, the objective is to reduce the number of items and the overall memory size according to [27], there is the feature separateness Loss defined with a margin of $\epsilon$ as follows:

$$\mathcal{L}_{sep} = \sum_{P}^{P}\left[\left\|\mathbf{W}_p^r - \mathbf{m}^{st}\right\|_2 - \left\|\mathbf{W}_p^r - \mathbf{m}^{nd}\right\|_2 + \epsilon\right]_+, \qquad (11)$$

where $P$ represents the number of queries, and $\mathbf{m}^{st}$ and $\mathbf{m}^{nd}$ represent the first and second nearest items for the query $\mathbf{W}_p^r$. Thus,

for the three memories, the separateness loss is denoted as $\mathcal{L}_{sep}^b$, $\mathcal{L}_{sep}^s$, and $\mathcal{L}_{sep}^r$ respectively. In summary, the overall loss function is expressed as:

$$\mathcal{L} = \mathcal{L}_{mo} + \alpha\mathcal{L}_{app} + \beta(\mathcal{L}_{sep}^b + \mathcal{L}_{sep}^s + \mathcal{L}_{sep}^r), \quad (12)$$

where $\alpha$ and $\beta$ are balancing hyper-parameters.

**Anomaly Score:** Measuring the anomaly scores involves three components: motion anomaly score, appearance anomaly score, and scene-related anomaly score. In the first level of our framework, we obtain high-level skeleton feature $f_{sk}$ and behavior feature $f_{rgb}$. Due to the distinct structures and input data of the two modules, When encountering previously unseen anomalous behaviors, there is a substantial difference between them, We can utilize Euclidean distance as the Motion Anomaly Score:

$$\mathcal{S}_{mo} = ||f_{sk} - f_{rgb}||. \quad (13)$$

Furthermore, as shown in Figure. 4(b), given the adoption of a 50% masking approach, there are two sets of mutually exclusive masks $MASK$ and $\overline{MASK}$ to ensure complete coverage for all patches. Therefore, the calculation method for appearance anomaly scores $\mathcal{S}_{app}$ is:

$$\mathcal{S}_{app} = \frac{1}{2}sim(f_u, f_m^1) + \frac{1}{2}sim(f_u, f_m^2), \quad (14)$$

where $sim(f_u, f_*) = 1 - f_u \cdot f_*/(||f_u|| \cdot ||f_*||)$. Next, we can determine scene-related anomalies based on the difference in matching weights $C_b$ and $C_s$ between behavior and scene:

$$S_{mm} = ||C_b - C_s||. \quad (15)$$

Taking all the above into consideration, the anomaly score for behavior can be expressed as:

$$Score = S_{mo} + \lambda_{app}S_{app} + \lambda_{mm}S_{mm}, \quad (16)$$

where $\lambda_{app}$ and $\lambda_{mm}$ are balancing hyperparameters. Finally, the scores are normalized to the range of 0-1 using min-max scaling. We employ the overlap sampling method, where the score of each video clip in a segment is used as the frame-level score for the intermediate frames.

## 4 EXPERIMENTS

### 4.1 Datasets

In this study, we evaluate the performance of our proposed method on two public datasets.

**ShanghaiTech** Dataset [24] is one of the largest datasets for video anomaly detection. The total number of frames used for training and testing reached 274K and 42K, respectively. It contains 13 scenes, a total of 330 test sets containing only normal event training videos and 107 normal and abnormal events, and is annotated at the frame and pixel levels. Some abnormal types of the dataset include: fighting, running in inappropriate scenes, cycling, etc.

**UBnormal** Dataset [1] is a newly proposed supervised multi-scene large-scale video anomaly detection dataset. The data set is generated using virtual animated characters and objects in Cinema 4D. It contains 268 training videos and 211 test videos. The data set has diverse scenes, diverse clothing and complex environments, such as insufficient light at night, foggy days, fires and so on. Abnormal behavior, including fighting, falling, lying down, running,

**Table 1: Comparison of AUC(%) Performance on Shang-haiTech(SHT) and UBnormal(UBN) datasets. We additionally list the input features of the approach, where $f_{rgb}$ denotes appearance, $f_o$ denotes optical flow and $f_{sk}$ denotes pose data. R, P, O represent reconstruction, prediction, and other methods, respectively. (*) indicates reproduction results. The best result (bold), and the second-best result (underlined).**

|    | Algorithm | Type | Feature | SHT | UBN |
|----|-----------|------|---------|-----|-----|
| **WE** | MIL [30] | - | $f_{rgb}$ | 85.3 | 62.1 |
|    | MIST [13] | - | $f_{rgb}$ | 94.8 | 68.2 |
|    | RTFM [34] | - | $f_{rgb}$ | 97.2 | 69.4 |
|    | MSL [20] | - | $f_{rgb}$ | 95.5 | - |
|    | LSTC [33] | - | $f_{rgb}$ | 97.8 | 77.5 |
| **UN** | MPEDRNN [26] | R | $f_{rgb}$ | 77.1 | 60.6 |
|    | GPEC [25] | R | $f_{rgb}$ | 76.1 | 53.4 |
|    | TimeSformer [6] | R | $f_{rgb}$ | - | 69.8* |
|    | DPU [39] | R | $f_o$ | 77.8 | - |
|    | AED [16] | R | $f_{rgb}$ | 82.7 | 59.3 |
|    | SSPCAB [28] | R | $f_{rgb}$ | 83.6 | - |
|    | COSKAD [14] | R | $f_{sk}$ | 75.6 | 65.5 |
|    | HSC [32] | R | $f_{rgb} + f_{sk}$ | 83.4 | - |
|    | USTNDSC [46] | P | $f_{rgb}$ | 73.8 | - |
|    | AMFT [7] | P | $f_{rgb} + f_o$ | 73.7 | - |
|    | AMSRC [18] | P | $f_{rgb} + f_o$ | 76.6 | - |
|    | SSMTL++ [4] | P+R | $f_{rgb} + f_o$ | 83.8 | 62.1 |
|    | HF$^2$-VAD [23] | P+R | $f_{rgb} + f_o$ | 76.2 | 65.2 |
|    | STG-NF [17] | O | $f_{sk}$ | 85.9 | 69.7* |
|    | JIGSAW [36] | O | $f_{rgb}$ | 84.3 | 56.4* |
|    | **Ours** | O | $f_{rgb} + f_{sk}$ | **86.9** | **74.3** |

crossing the road and so on, is a very challenging data set. Following the approach of [17, 33], we remove anomalous video segments from the training set, retaining only normal video segments for model training.

### 4.2 Implementation Details

All experiments are conducted using PyTorch on an NVIDIA RTX 3090. The Normalizing Flow is pre-trained using a flow number of $L = 8$. For the training of RGB Encoder and Mask Encoder, the SGD optimizer with a learning rate of 0.0001 and a momentum of 0.5 is utilized. Behavior, Scene, and Behavior-Scene Matching Memory each have 128, 32, and 64 memory items, respectively. The pre-trained model $\mathcal{F}_B$ utilized the representation layers from the only layers in first two stages of the tiny video Swin Transformer model trained on the Kinetics-400 dataset. In the final loss function $\mathcal{L}$, the parameter $\alpha$ is set to 0.1 and $\beta$ to 0.001. In anomaly score $Score$, both $\lambda_{app}$ and $\lambda_{mm}$ are set to 0.1. The parametric experiments can be found in the Appendix.

### 4.3 Evaluation Metric

Following [17, 23, 36], we use the AUC(area under the ROC curve) at the frame level as the evaluation metric, which measures the relationship between true positive rate and false positive rate, used

**Table 2: The AUPRC and Max-F1(%) performance of different types of measurement methods..**

| Method | SHT | | UBN | |
|---|---|---|---|---|
| | AUPRC | Max-F1 | AUPRC | Max-F1 |
| ROADMAP[38] | 72.4 | 74.5 | - | - |
| HF$^2$-VAD[23] | 71.8 | 79.8 | - | - |
| JIGSAW[36] | 73.7 | 82.3 | 60.3 | 67.7 |
| STG-NF[17] | 85.9 | 82.2 | 63.1 | 70.1 |
| Our | **88.0** | **83.5** | **65.3** | **70.5** |

to assess the model's performance at different thresholds. The AUC score is calculated by connecting all frames and computing the micro-averaged AUC score [17]. Futhermore, in real-world scenarios, abnormal events are generally less frequent than normal events. We also use AUPRC(Area Under the Precision-Recall Curve) and Max-F1 to evaluate the model's performance. The higher the scores of these three metrics, the better the performance of anomaly detection.

## 4.4 Comparison with Other Methods

As depicted in Tab. 1 We present an AUC comparison of our method with cutting-edge video anomaly detection methods that have been published in recent years, including both some weakly supervised(WE) and unsupervised(UN) learning-based approaches. By analyzing the table, it can be observed that our method achieves impressive AUC scores of 86.9% on the ShanghaiTech dataset(SHT) and 74.3% on the UBnormal dataset(UBN). These scores surpass the previous state-of-the-art unsupervised learning method by 1.16% and 6.45%, respectively. Notably, our method also outperforms most of the weakly supervised learning methods on the UBnormal dataset. These results confirm the effectiveness of our proposed Multilevel Guidance-Exploration framework for human behavior anomaly detection tasks.

As depicted in Tab.2, we further augment our evaluation by comparing AUPRC and max-F1 metrics across four algorithms: the existing algorithm ROADMAP[38], currently recognized for having the highest publicly known AUPRC performance, and three open-source methods, including HF$^2$-VAD[23], JIGSAW[36], and STG-NF[17]. Our algorithm outperforms others on both datasets, particularly excelling in the AUPRC metric, surpassing the second-best algorithms by 2.4% and 3.2% respectively. This indicates that our model maintains high precision at different recall rates and minimizes false positives when capturing anomalies, further underscoring the effectiveness of the method.

## 4.5 Visualization Evaluation

The anomaly scores in two distinct scenes from the ShanghaiTech and UBnormal datasets are visualized in Figure 5. The figure displays the results of our final method. The red area represents the time interval in which the ground-truth anomaly occurs.

*Appearance Anomaly*: Figure 5 (1) depicts an anomaly where a person is riding an electric scooter on a pedestrian path. Although "sitting" appears to be a relatively normal posture, a high anomaly

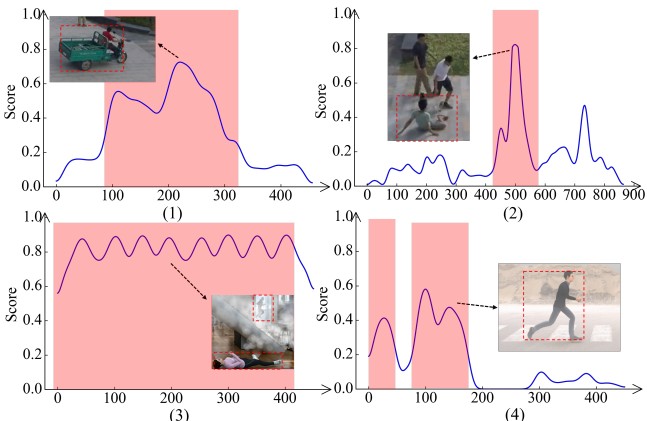

**Figure 5: Visualizing Anomaly Scores for Video Frames** $01-0053$, $04-0004$ **from the ShanghaiTech Dataset, and** *abnormal$-$scene$-5-2-$smoke*, *abnormal$-$scene$-11-2-f$og* **from the UBnormal Dataset.**

score is obtained due to unexpected appearance detection. Our method effectively detects this appearance anomaly.

*Motion Anomaly and Environment Disruption*: Figures 5 (2), (3) and (4) illustrate motion anomalies, specifically fighting, lying, and jaywalking. Particularly, the behavior of lying down is challenging for many algorithms to detect, but our algorithm can capture these motion anomalies. Additionally, our algorithm performs well in smoky or foggy conditions, which emphasizes the robustness of our approach in handling variations in environmental disruption.

## 4.6 Ablation Studies

We first conducted ablation experiments on modules and loss functions, followed by ablation experiments on measurement methods.

**Ablation Study on Components and Loss Functions**: We analyze the contribution of each component and loss function and report the results in Table 3. From the first group of ablation experiments, we find that the LSTA module significantly improves model performance, while the BSMM module, is less impactful. This is attributed to the dataset's design, which is behavior and scene-independent in UBnormal datasets. Further experiments in Section 4.6 validate the effectiveness of the BSMM module. From the second group of experiments, we observe that both proposed loss functions are effective, with $L_{mo}$ notably enhancing performance. This is primarily due to the fact that anomalies in human behavior are frequently associated with motion. The improvement in performance validates the effectiveness of our guidance-exploration pattern.

**Feature Combination for Computing Anomaly Score:** We expound on our measurement methodology in Sections 1 and 3. However, does leveraging a single network output yield superior outcomes compared to employing measurement techniques involving two networks? To ascertain the superiority of our methods, we experimentally compare performance disparities among various methodologies.

**Table 3: Ablation experiments on modules and loss function.**

| Group | LSTA | BSMM | $L_{mo}$ | $L_{app}$ | SHT | UBN |
|-------|------|------|----------|-----------|-----|-----|
|       |      |      | ✓        | ✓         | 84.9 | 69.6 |
| **1** | ✓    |      | ✓        | ✓         | 86.5 | 73.9 |
|       |      | ✓    | ✓        | ✓         | 85.1 | 69.4 |
|       | ✓    | ✓    | ✓        |           | 86.1 | 73.9 |
| **2** | ✓    | ✓    |          | ✓         | 78.3 | 67.2 |
|       | ✓    | ✓    | ✓        | ✓         | **86.9** | **74.3** |

**Table 4: The AUC(%) performance of different types of measurement methods. Thenotation** $OCSVM(f_*)$ **refers to the method of feeding feature** $f$ **into a one-class SVM for training and obtaining anomaly scores.** $f_G$ **represents** $f_{sk}$ **in 1-st and** $f_{un}$ **in 2-nd level.** $f_E$ **represents** $f_{rgb}$ **in 1-st and** $f_m$ **in 2-nd level framework(As shown in Fig.2). Noted that this experiment disregarded the impact of BMSS on the experiment.**

| Method | 1-st level | | 2-nd level | |
|--------|------------|------|------------|------|
|        | SHT | UBN | SHT | UBN |
| $OCSVM(f_G)$ | 83.8 | 64.5 | 85.3 | 72.7 |
| $OCSVM(f_E)$ | 81.2 | 66.5 | 86.2 | 72.9 |
| $\|f_G + f_E\|$ | 67.7 | 50.7 | 85.9 | 72.6 |
| $\|f_G - f_E\|$ | **86.5** | **73.9** | 86.3 | 73.6 |
| $sim(f_G, f_E)$ | 85.7 | 72.3 | **86.5** | **73.9** |

As illustrated in Table 4, we conduct experiments to validate the effectiveness of our 1-st and 2-nd level guidance-exploration pattern and anomaly measurement methods. The results indicate that both cosine similarity and Euclidean distance, which can reflect distance difference, have achieved excellent outcomes, far surpassing other measurement methods. Furthermore, to depict the distribution of anomalies and normal samples, we take the 1st-level Guidance Exploration as an example, and we plot histograms in Figure 6 for both our method($\|f_{sk} - f_{rgb}\|$) and well-performing measurement method that only use $f_{sk}$ generated by Normalizing Flow($OCSVM(f_{sk})$). Our approach assigns higher scores to anomalies, highlighting the effectiveness of our guidance-exploration pattern.

## 4.7 Explore the Capability of the Behavior-Scene Matching Memory

In existing datasets, anomalies related to scenes are relatively rare. Similar to [32], we adapt the dataset and perform the following experiments to assess the effectiveness of the Behavior-Scene Matching Method.

Initially, within the UBNormal dataset, the actions of "running" and "jumping" were deemed abnormal across all scenes. We consider these two actions as normal in the appropriate scenes. Firstly, we extract a subset from the UBNormal dataset, encompassing Scenes 1, 3, 4, 5, 11, 20 and 21. Subsequently, we categorize these scenes into four types: Outdoor street scenes A, Indoor station scenes B, Outdoor zebra crossing scenes C, and Indoor office scenes D. Following that, we posit that "running" and "jumping" are considered

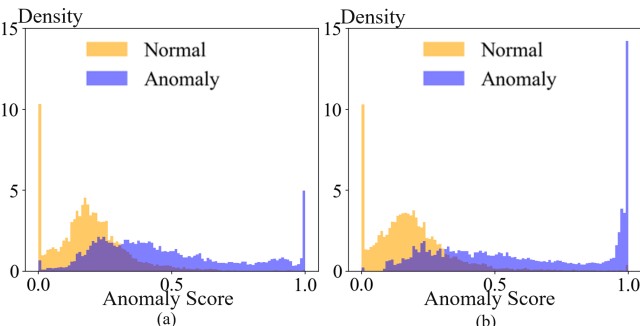

**Figure 6: Histograms of anomaly scores for abnormal and normal data. (a) Normalizing Flow, (b) Our Guidance-Exploration Network in 1-st level.**

**Table 5: The AUC(%) performance of different methods. Here, A, B, C, and D are datasets derived from different scenes.**

| Method | [A,C,D] | [B,C,D] | [A,B,C,D] | Origin |
|--------|---------|---------|-----------|--------|
| Jigsaw [36] | 58.2 | 59.7 | 61.0 | 57.1 |
| STG-NF [17] | 62.1 | 67.3 | 65.8 | 65.9 |
| HF$^2$-VAD [23] | 59.2 | 58.0 | 59.9 | 62.4 |
| **Our w/o BSMM** | 63.5 | 66.8 | 65.1 | **67.4** |
| **Our** | **64.1** | **68.2** | **67.1** | 67.2 |

normal within scenes A and B. Subsequently, videos containing these behaviors and no other anomalies are moved from the test set to the training set. Finally, as shown in Table 5, we divide the mentioned data into three settings for both training and testing. Concurrently, we use the data from the original nine scenes as a control group.

As depicted in Table 5, our approach demonstrates superior performance improvement across the three experimental sets compared to the scenario where the Behavior-Scene Matching Memory is not employed. Furthermore, our method surpasses advanced methods with publicly accessible code, validating the effectiveness of our Behavior-Scene Matching Memory in identifying anomalies related to scenes.

## 5 CONCLUSION

We introduce a pioneering framework named the Multilevel Guidance-Exploration Network, along with a behavior-scene matching approach, which not only improves the accuracy of detecting motion and appearance anomalies but also identifies scene-related behavioral anomalies across various scenarios, significantly expanding its applicability. Experimental results confirm the method's effectiveness. Furthermore, to ensure swift inference speed, we keep the parameter count of these models low. However, the model does have certain drawbacks. It is non-end-to-end, making it relatively complex, and the training process is somewhat cumbersome. Nevertheless, the actual trainable parameter count remains modest.

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
