# OpenReview forum: "A Multilevel Guidance-Exploration Network and Behavior-Scene Matching Method for Human Behavior Anomaly Detection"
_acmmm.org/ACMMM/2024/Conference — MM2024 Poster_

### Official Review · Reviewer_rGAQ · 2024-05-14

**Rating:** 4
**Confidence:** 4

**Summary:**

The manuscript proposes a novel framework called Multi-level Guided Exploration Network (MGENet), which detects anomalies through differences in high-level representations between the guidance network and the exploration network. The method utilizes a normalization process with skeletal keypoints as input to guide an RGB encoder (with unmasked RGB frames as input) to explore underlying motion features. The RGB encoder then guides the mask encoder (taking the masked RGB frames as input) to explore the underlying appearance features. Furthermore, we design a behavioral scenario matching module (BSMM) to detect scenario-related behavioral anomalies.

**Strengths:**

1.The method proposed in the manuscript outperforms existing unsupervised methods.
2.This manuscript focuses on the connection between scene and action, which is an interesting idea.

**Limitations:**

1.The manuscript mentions that the proposed method can identify behavioural anomalies in various scenarios that are relevant to the scenario, can this be better demonstrated through visualisation? 2.The proposed method is relatively complex and it is desirable to provide the number of training parameters and inference time and compare it with other methods. 3.So far, there have been some methods that focus on scene information, behavior information, or both at the same time, the author should compare the performance and method level with such works published on ACM MM:

[1] Cloze test helps: Effective video anomaly detection via learning to complete video events[C]//Proceedings of the 28th ACM international conference on multimedia. 2020: 583-591.

[2] Hierarchical scene normality-binding modeling for anomaly detection in surveillance videos[C]//Proceedings of the 30th ACM international conference on multimedia. 2022: 6103-6112.

[3] Scene-aware context reasoning for unsupervised abnormal event detection in videos[C]//Proceedings of the 28th ACM international conference on multimedia. 2020: 184-192.

**Suitability:**

3

---

### Official Review · Reviewer_s6m3 · 2024-05-24

**Rating:** 4
**Confidence:** 3

**Summary:**

The paper presents a framework termed the Multilevel Guidance-Exploration Network (MGENet) for the detection of human behavior anomalies. The approach cleverly integrates several existing methodologies to achieve improved performance in anomaly detection. The authors propose a system that leverages both motion and appearance features, guided by a Normalizing Flow model, to detect anomalies in video sequences. Additionally, they introduce a Behavior-Scene Matching Module (BSMM) to identify scene-related anomalies. The paper reports commendable results on multiple datasets, demonstrating the effectiveness of the proposed method. However, the novelty of the approach seems somewhat limited as it appears to be an amalgamation of existing techniques.

**Strengths:**

1. The integration of multiple modalities (motion, appearance, and scene context) for anomaly detection is a strength that contributes to the robustness of the framework.
2. The method has been validated on multiple datasets, which lends credibility to the authors' claims of its effectiveness.
3. The visualization of results provides a clear demonstration of the method's ability to detect anomalies in various scenarios.

**Limitations:**

1. While the fusion of existing methods yields positive results, the novelty of the approach is not strong. The paper could benefit from a more explicit discussion on how this work differs from or improves upon existing methods.
2. There is a concern regarding the generalizability of the motion anomaly detection component, which seems to rely on skeletal keypoints and may not be equipped to handle scenarios with multiple individuals in the frame. This limitation is significant and could reduce the practical applicability of the method.
3. The non-end-to-end nature of the model and the complexity of the training process could be potential barriers to adoption, particularly in real-world applications.
4. The paper could provide more insight into how the method performs in terms of computational efficiency, which is critical for real-time anomaly detection systems.

**Suitability:**

3

---

### Official Review · Reviewer_sgwN · 2024-05-26

**Rating:** 2
**Confidence:** 3

**Summary:**

The authors propose a novel framework called the Multilevel Guidance-Exploration Network (MGENet), which detects anomalies through the difference in high-level representation between the Guidance and Exploration network. Specifically, they first utilize the Normalizing Flow that takes skeletal keypoints as input to guide an RGB encoder, which takes unmasked RGB frames as input, to explore latent motion features. Then, the RGB encoder guides the mask encoder, which takes masked RGB frames as input, to explore the latent appearance feature. Additionally, they design a Behavior-Scene Matching Module (BSMM) to detect scene-related behavioral anomalies. Extensive experiments demonstrate that their proposed method achieves state-of-the-art performance on ShanghaiTech and UBnormal datasets, with AUC of 86.9 % and 74.3 %, respectively.

**Strengths:**

The experiments are comprehensive, and the motivation is clear.

**Limitations:**

The description of the method is very unclear, and many symbols are not well-defined, resulting in poor readability of the paper. Additionally, the method lacks innovation.

**Suitability:**

2

---

### Meta-Review · Area_Chair_cBPt · 2024-06-30

**Recommendation:** Accept (Poster)
**Confidence:** 5

**Metareview:**

This work is focused on detecting anomalies in human behavior. The authors have explored different modalities derived from RGB videos to solve this task. This work initially received weak reject and 2x borderline accept ratings. The main concerns were lack of novel ideas (Reviewer sgwN,s6m3), compute and parameter analysis (Reviewer s6m3 and rGAQ), generalizaton (s6m3), comparison with more methods (rGAQ). The authors provided a rebuttal and most of the concerns were addressed. The final ratings are weak accept, borderline accept and weak reject. One of the remaining concern is about 'limited innovation and poor writing' from Reviewer sgwN. Although this work utilize several existing ideas, but their integration is non-trivial. After considering both the weakness and strengths of this work, the AC recommend this work to be accepted as it has some interesting novel aspects/ideas and the ideas are validated with extensive set of experiments with good performance. The authors are advised to consider the reviews about writing and improve it in the final submission.